# Quantum Mutual Information, Fragile Systems and Emergence

**DOI:** 10.3390/e24111676

**Published:** 2022-11-17

**Authors:** Yasmín Navarrete, Sergio Davis

**Affiliations:** 1Instituto de Filosofía y Ciencias de la Complejidad, Los Alerces 3024 Ñuñoa, Santiago 7780192, Chile; 2Research Center in the Intersection of Plasma Physics, Matter and Complexity (P^2^mc), Comisión Chilena de Energía Nuclear, Casilla 188-D, Santiago 8320000, Chile; 3Departamento de Física, Facultad de Ciencias Exactas, Universidad Andres Bello, Sazié 2212 piso 7, Santiago 8370136, Chile

**Keywords:** emergence, quantum theory, density matrix, subadditivity

## Abstract

In this paper, we present an analytical description of emergence from the density matrix framework as a state of knowledge of the system, and its generalized probability formulation. This description is based on the idea of fragile systems, wherein the observer modifies the system by the measurement (i.e., the *observer effect*) in order to detect possible emergent behavior. We propose the use of a descriptor, based on quantum mutual information, to calculate if subsystems of systems have inner correlations. This may contribute to a definition of emergent systems in terms of emergent information.

## 1. Introduction

From quantum mechanics, it is well known that any separable state belongs to a composite space that can be factored into individual states from separate subspaces. One state is said to be correlated if it is not separable. As a matter of fact, to determine if a state is separable or not is not trivial and the problem is classed as *NP-hard* from the theory of complex systems [1].

A density operator, in quantum mechanics, is used to describe the statistical state of a quantum system. The usual meaning of it is that the eigenvalues are the probabilities of finding a system in one state corresponding to the eigenvectors. In a physical sense, we can see its elements as relative frequencies corresponding to an appropriate ensemble of *N* identical copies of the system that are in several possible states under a certain setup or preparation protocol. Thus, we have a superposition of quantum states $|v_{i}\rangle$ with probabilities $p_{i}$ (real numbers) satisfying $\sum p_{i}=1$. However, it can be shown that the density operator formalism can be recovered in a Bayesian formalism for noncommutative expectations, wherein the system depends on the order of the measurements. This is not restricted just to a quantum mechanical system, and can be understood under the framework of what we have called a *fragile system* [2]. This operator provides a convenient mean for describing quantum systems whose state is not completely known, it being mathematically equivalent to a state vector approach [3,4].

By applying the entropy to the density matrix, we can obtain the degree of disinformation of the state of the system. The systems can be composed of subsystems and, using the subadditivity property (the probability of the whole is less than that of its parts) [5], it is possible to quantify if the entropy of the whole is less than that of its parts. Holzer and De Meer [6] make a comparison between the information at the system level with the information at a lower level. As they state, “this measure gives a high value of emergence for systems with many dependencies (interdependent components) and a low value of emergence for systems with few dependencies (independent components)”; therefore, the information of the whole is more than the information of the parts. In that sense, the entropy can be a good parameter to measure a type of emergence in systems.

This paper is organized as follows. In Section 2, we talk about emergent systems and its current definitions. In Section 3, we define fragile systems as ones that are modified by the act of measurement because of the change in internal variables. In Section 4, we introduce the density matrix formalism. In Section 5, we depict a mathematical formulation of emergent systems within the density matrix formalism. In Section 6, we show a concrete example of a subadditive system. Finally, we provide some concluding remarks in Section 7.

## 2. Emergent Systems

Several definitions of emergence exist, taking into account different aspects of their origin or behavior. For instance, Peter Checkland [7] defines emergence as “the principle that entities exhibit properties which are meaningful only when attributed to the whole, not to its parts”. Emergent systems are structured in such a way that their components interact, allowing for the structure of global patterns, depicted as a consequence of interrelations/correlations between subsystem elements, them being the result of complex and self-organizing processes. This process may be triggered by an external stimulus.

There are basically three types of emergence: simple, weak and strong ones, described as follows:

*Simple emergence* is composed by the combination of certain properties and relationships between elements in a non-linear behavior. For instance, in order to achieve the flight of an airplane, we cannot consider the motors, the propulsion system and their wings separately; all of these properties must be considered together because they are interconnected and they have interrelations through which flight emerges. This type of emergence can be predicted from the functioning of its parts and it is referred to as the concept of synergy, which means interacting or working together [8].

*Weak emergence* describes the emergence of properties of systems that may be predictable (not completely) and also reducible. They can be reduced to basic rules at an initial time. After a while, the behavior can be unpredictable, as is mentioned in chaos theory [9]; nevertheless, it is possible to make computational simulations about such systems because of the knowledge of the basic rules. Weak emergence is the product of complex system dynamics (i.e., non-linear behavior, spontaneous order and adaptation); an example of the latter is cellular automaton, known as Conway’s Game of Life [10,11,12].

*Strong emergence* is a case of non-expected emergence, as well as weak emergence. The difference lies in its non-reducible behavior, which appears just when the system is running. As it is systematically determined by low-level attributes, it is not possible to deduce it from the components at lower levels. The consciousness phenomenon is one example of this type of emergence, and appears as a construction process. There is likely no algorithm from the bottom up because it is a dynamical process evolving along time at the highest level, with non-linear relations at the lower ones.

Another way of conceptualizing emergence is the separation of levels of complexity of the system at different spatial or temporal scales [13].

Some main characteristics of strong emergence to consider are the following:**Non-reducible phenomenon**: the global state of the emergent system cannot be explainable, and neither is it reducible to its sub-system components.**Downward causation**: emergent high-level properties appear from a non-obvious consequence of low-level properties, but, at the same time, all processes at the lower level of hierarchy are constrained by and act in coherence with the laws of the higher level [14].**Wholeness**: a phenomenon wherein a complex, interesting high-level function appears as a result of combining low-level mechanisms in straightforward ways.**Radical novelty emergence**: a phenomenon wherein a system is designed according to certain principles. Interesting unexpected properties arise from the behavior of sub-system elements [15].

In general, emergent systems are common in nature and technology. One example of the latter is the speed of a vehicle affected by the center of gravity, the driver skills, weather and friction, among other attributes (in this case, we can predict the emergent property of the speed from the relation between components, it being the case of weak emergence). It is possible to find strong emergent properties in the consciousness phenomenon [16], human body and social phenomena, among others. One important example is ’self-awareness’, which is a result of the interconnection of neurons in the brain [17].

In the case of biological systems (as well as in the case of social phenomena), emergent models are appropriate for describing those situations. We can observe the characteristics mentioned regarding strong emergence: non-reducibility, radical novelty, wholeness and downward causation.

The main aspect of biological emergent systems is that they may be observed from inside them by the system (“from internal control of a system which might be fully controlled by an observer/controller architecture that is part of the system” [18]); this is in concordance with the autopoiesis theory proposed by the Chilean biologists H. Maturana and F. Varela [19] to define the self-maintaining chemistry of any living cell, which is under the perspective of chemical organization theory used to formalize autopoietic structures, “providing a basis to operationalize goals as an emergent process” [20]. When the measurement is affected by the observer, it is the case of what we will call a *fragile system*. In the next section, we will go deeper into the concept.

## 3. Fragile Systems

In simple terms, a fragile system is one that is affected by the measurement [21]. This distinguishes it from a non-fragile (classical) system, which is not modified upon observation.

Because any system (being fragile or not) possesses information, we will think of a system as a “black box” that can be found in different internal states, which is to be denoted by λ. In general, λ contains many degrees of freedom, but we will not make use of that inner structure here. The internal state λ contains all of the information necessary to describe any aspect of the system.

The crucial difference between a fragile system and a non-fragile one is that, in a fragile system, access to the internal state λ is impossible, because it is precisely this internal state that is modified by the measurement. As the modification of the state λ depends on the details of the environment when carrying out the measurement (which we do not know or control with accuracy), the outcome of a measurement is unavoidably stochastic, and a mathematical formulation requires probability theory [22].

## 4. The Density Operator

In this section, we formulate the density matrix operator, as a previous concept, in order to define emergent systems using the calculation of entropies.

The density operator is a positive semi-definite Hermitian operator of trace one. If A is the matrix representation of an arbitrary observable $\hat{A}$, we can write
(1)$$\begin{array}{ccc} A & = & \left(\begin{array}{c} a_{1}\\ a_{2}\\ \vdots \\ a_{N} \end{array}\right)\left(\begin{array}{c} {\bar{a}}_{1}{\bar{a}}_{2}\cdots {\bar{a}}_{N} \end{array}\right). \end{array}$$
Hence, we have
$$\begin{array}{ccc} A & = & \left(\begin{array}{ccc} a_{1}{\bar{a}}_{1} & \cdots & a_{1}{\bar{a}}_{N}\\ a_{2}{\bar{a}}_{1} & \cdots & a_{2}{\bar{a}}_{N}\\ \vdots & \ddots & \vdots \\ a_{N}{\bar{a}}_{1} & \cdots & a_{N}{\bar{a}}_{N} \end{array}\right) \end{array}$$
where ${\bar{a}}_{i}$ is the conjugate of the element $a_{i}$, and, in the case of real numbers, the same element. We can take the average of different measurements represented by different matrices A of the same observable, and then normalize and diagonalize it; finally, we obtain the density matrix of a mixture as below: (2)$$\hat{\rho }=\sum_{i}^{N}p_{i}|v\rangle \langle v|$$
where $p_{i}=P\left(\lambda_{i}|I\right)$ and $\lambda_{i}$ is obtained from the eigen-value problem $\lambda_{i}|v_{i}\rangle =\lambda_{i}\hat{A}$ [21].

We can formulate the density matrix operator by the use of a complex Hilbert space just as in Von Neumann’s formulation of quantum theory [23]. For this, we consider an arbitrary orthonormal basis set |*n*〉 (*n* = 1, …, *N*) with 〈i|j〉=δ(i,j), and define the density operator [24] $\hat{\rho }$ as
(3)$$\hat{\rho }:=\sum_{i=1}^{N}\sum_{j=1}^{N}\rho_{ij}|i\rangle \langle j|,$$
with $\rho_{ij}$ complex numbers. Imposing that $\hat{\rho }$ is Hermitian, we see that the diagonal elements $\rho_{ii}$ must be real and $\rho_{ij}=\bar{\rho_{ji}}$. It is always possible to make a choice of such complex matrix elements $\rho_{ij}$ so that they are proportional to the elements $p_{ij}$; these are given by
(4)$$\rho_{ij}=\langle \epsilon_{i}|\hat{\rho }|\epsilon_{j}\rangle =\frac{1}{2\eta }\left(\left[p_{ij}+p_{ji}\right]+i\;\left[p_{ij}-p_{ji}\right]\right),$$
where $\eta =\sum_{i}p_{ii}$ is a normalization factor that imposes $\mathrm{Tr}(\hat{\rho })=1$.

### Pure and Mixed States

Consider an ensemble of measurements ${\left\{|v_{i}\rangle \right\}}_{i=1}^{N}$. If the state vector is known [25], the ensemble represents a pure state. Assuming that the system is in the state |*v*〉, we can expand it with respect to the eigenvector of a Hermitian operator $\hat{A}$ as follows: (5)$$\begin{array}{ccc} |v\rangle & = & \sum_{n}\lambda_{n}|v_{n}\rangle , \end{array}$$(6)$$\begin{array}{ccc} \hat{A}|v\rangle & = & \lambda_{n}|v\rangle . \end{array}$$

Finally, we can define a pure state by the following term [25]: (7)$$\hat{\rho }=|v\rangle \langle v|.$$

Because ${\hat{\rho }}^{2}=\left(|v\rangle \langle v|\right)\left(|v\rangle \langle v|\right)=|v\rangle \left(\langle v|v\rangle \right)\langle v|=\hat{\rho }$, we can distinguish a density operator of a pure state by tracing; then, we have
(8)$$\mathrm{Tr}(\hat{\rho })=\mathrm{Tr}\left({\hat{\rho }}^{2}\right)=1.$$

When we cannot repeat exactly the same initial condition, because of the noise of the system, we represent this situation in a mathematical formulation in terms of an operator called a density matrix for mixed states. This is a superposition of pure states [26]: (9)$$\hat{\rho }=\sum_{i}p_{i}|v\rangle \langle v|,$$
where the weights of each measurement satisfy the normalization condition $\sum_{i}p_{i}=1$. Each $p_{i}$ is the probability of finding a system in a given pure state.

In contrast to a pure state, when we have a mixed-density matrix, the trace of the square density matrix is given by the inequality
(10)$$\begin{array}{c} \mathrm{Tr}({\hat{\rho }}^{2})<1. \end{array}$$

Hence, Tr $\left({\hat{\rho }}^{2}\right)$, known as the *degree of purity* [25], can be used to distinguish between pure and mixed states in a basis-independent manner.

## 5. Density Operator Formalism and Emergent Systems

Earlier work by Prokopenko et al. [13] has set the basis for a discussion of complexity, self-organization and the emergence of classical systems in information-theoretical terms, particularly in terms of Shannon entropy and mutual information.

Taking all of this into account, we will rephrase the earlier arguments in terms of the (von Neumann) entropy of density matrices. However, first, let us review the behavior of the information entropy (or Shannon entropy) for classical, correlated systems.

We adopt the Bayesian view by Jaynes [27] and others of the Shannon entropy related to the information content of a model based on, in principle, subjective probabilities, but consistent with known facts. Shannon entropy is then a measure of missing information in a probabilistic model about some aspect of reality, and is therefore dependent of the state of knowledge used to construct said model. For a state of knowledge I, where we ask an arbitrary question with *N* possible answers, denoted by the proposition $A_{1},A_{2},\ldots ,A_{N}$, the Shannon entropy is defined as
(11)$$S\left(I\right):=-\sum_{i=1}^{N}p_{i}\ln p_{i}$$
where $p_{i}$ is the probability $P(A_{i}|I)$ of the answer $A_{i}$ being true under I. Please note that, for two ‘observers’ with different states of knowledge $I_{1}$ and $I_{2}$, the Shannon entropies $S(I_{1})$ and $S(I_{2})$ that they assign to an unknown question will, in general, be different. For instance, if the first observer knows that $A_{1}$ is true, whereas the second only knows that either $A_{1}$ or $A_{2}$ is true, then $S(I_{1})=0$ because $P\left(A_{k}|I_{1}\right)=\delta_{1k}$, but $S(I_{2})>0$ because $0<P(A_{1}|I_{2})<1$ and $P\left(A_{2}|I_{2}\right)=1-P\left(A_{1}|I_{2}\right)>0$.

In the case where the question involves the unknown value of one or more variable, the information entropy directly translates in terms of the probability distribution. For instance, for the joint probability distribution P(X,Y|I) of the variables *X* and *Y* under the state of knowledge I, we have
(12)$$S_{XY}\left(I\right):=-\sum_{x}\sum_{y}P\left(x,y|I\right)\ln P\left(x,y|I\right).$$

Using the product rule of probability,
(13)P(x,y|I)=P(y|x,I)P(x|I),
this entropy can always be separated into two terms [28],
(14)$$S_{XY}\left(I\right)=S_{X}\left(I\right)+{\langle S_{Y|X}\rangle }_{I},$$
where the first term is the entropy of the variable *X*, and the second term,
(15)$${\langle S_{Y|X}\rangle }_{I}:=-\sum_{x}P\left(x|I\right)\sum_{y}P\left(y|x,I\right)\ln P\left(y|x,I\right)$$
is the expected value of the conditional entropy of *Y* given *X*. This conditional entropy cannot be negative, it being the expected value of a non-negative quantity.

It is possible to extend the Bayesian idea of probabilities as degrees of belief constrained on the available information to quantum systems [29,30]. Let $\rho_{12}=\rho_{1}⊗\rho_{2}$. We can obtain the eigenvalues of $\rho_{1}$$a_{i}={\left\{\rho_{1}\right\}}_{ii}$ and $\rho_{2}$$b_{j}={\left\{\rho_{2}\right\}}_{ii}$. Hence, the diagonal elements of $\rho_{12}$ are given by all products of the form $a_{i}b_{j}$, where $i=1,...,n_{1}$ and $j=1,...,n_{2}$. Here, $n_{1}$ and $n_{2}$ are the dimensions of $\rho_{1}$ and $\rho_{2}$, respectively; then: (16)$$\begin{array}{ccc} S(\rho_{12}) & = & -\mathrm{Tr}(\rho_{12}\ln\rho_{12})\\ & = & -\sum_{i=1}^{n_{1}}\sum_{j=1}^{n_{2}}\left(a_{i}b_{j}\right)\ln\left(a_{i}b_{j}\right). \end{array}$$

On the other hand, the sum of the entropy of each system is given by
(17)$$S\left(\rho_{1}\right)+S\left(\rho_{2}\right)=-\sum_{i=1}^{n_{1}}a_{i}\ln\left(a_{i}\right)-\sum_{j=1}^{n_{2}}b_{j}\ln\left(b_{j}\right).$$

Thus, the entropy for an ensemble $\left(\rho_{12}\right)$ for which the subsystems are uncorrelated is just equal to the sum of the entropies of the reduced ensembles for the subsystems. When there are correlations, we should expect an inequality instead (called the subadditivity property of entropy [5,31,32]), since, in this case, $\rho_{12}$ contains additional information concerning the correlations, which is not present in $\rho_{1}$ and in $\rho_{2}$ (those are the partial traces of $\rho_{12}$, respectively), as in
(18)$$S\left(\rho_{12}\right)\leq S\left(\rho_{1}\right)+S\left(\rho_{2}\right).$$

Given this inequality, we can use the mutual information formulation
(19)$$I:=S\left(\rho_{1}\right)+S\left(\rho_{2}\right)-S\left(\rho_{12}\right),$$
such that I>0 whenever there is subadditive behavior and I=0 for additivity.

From Equation (19), we obtain a descriptor of weak emergency in systems that are correlated. In these types of systems, new information emerges from the relation of its parts in terms of correlations.

## 6. An Example of a Subadditive System

As is depicted in Equation (4), we can write the density matrix as follows: (20)$$\rho_{ij}=\langle \epsilon_{i}|\hat{\rho }|\epsilon_{j}\rangle =\frac{1}{2\eta }\left(\left[p_{ij}+p_{ji}\right]+i\left[p_{ij}-p_{ji}\right]\right).$$
but where we now interpret $p_{ij}$ as
(21)$$\begin{array}{cc} p_{ij} & =P(\epsilon_{i},{\epsilon_{j}}^{\prime }|I), \end{array}$$
(22)$$\begin{array}{cc} \eta & =\sum_{i}p_{ii} \end{array}$$
where primed states are states after a measurement.

Because of the marginalization rule, it must hold that
(23)$$\sum_{i}p_{ij}=P\left(\epsilon_{j}^{\prime }|I\right)\;\mathrm{for}\;\mathrm{all}\;\;j=1,\ldots ,N,$$
while, simultaneously,
(24)$$\sum_{j}p_{ij}=P\left(\epsilon_{i}|I\right)\;\mathrm{for}\;\mathrm{all}\;\;i=1,\ldots ,N$$
must be true. As an example, consider an abstract system composed of two integers, *a* and *b*, such that a∈{1,2,3,4} and b∈{1,2,3,4}. Let us set the constraint I that *a* and *b* are either both even or both odd; then, there are eight allowed states, namely
(25)$$\begin{array}{cc} \Gamma_{1}: & \;a=1,b=1,\\ \Gamma_{2}: & \;a=1,b=3,\\ \Gamma_{3}: & \;a=3,b=1,\\ \Gamma_{4}: & \;a=3,b=3,\\ \Gamma_{5}: & \;a=2,b=2,\\ \Gamma_{6}: & \;a=2,b=4,\\ \Gamma_{7}: & \;a=4,b=2,\\ \Gamma_{8}: & \;a=4,b=4. \end{array}$$

Let us now define the *measurement* M(a,b) by
(26)$$M\left(a,b\right):=\left\{\begin{array}{c} 1\;\mathrm{if}\;a=b,\\ 2,\;\mathrm{if}\;a>b,\\ 3\;\mathrm{if}\;a<b. \end{array}\right.$$
such that the system undergoes the following transitions: $\Gamma_{1}↔\Gamma_{4}$, $\Gamma_{5}↔\Gamma_{8}$, $\Gamma_{2}↔\Gamma_{6}$ and $\Gamma_{3}↔\Gamma_{7}$. In this way, if the system has *a* = 3 and *b* = 1, that is, it is in the state $\Gamma_{3}$, and we perform the measurement *M*, we will obtain a value M=2 and the state will change to $a^{\prime }$ = 4 and $b^{\prime }$ = 2, that is, to $\Gamma_{7}$. This makes the system fragile regarding the measurement *M*, with the integers *a* and *b* being the *hidden variables* of the system. Lastly, we will define the parity of each integer as 0 if the number is odd, and 1 if it is even. In this way, there are only four observable states, namely
(27)$$\begin{array}{cc} s_{1} & =(0,0),\\ s_{2} & =(0,1),\\ s_{3} & =(1,0),\\ s_{4} & =(1,1), \end{array}$$
where the first and second position of the tuple represent the parity bit of *a* and *b*, respectively. Of these states, only $s_{1}$ and $s_{4}$ are consistent with the constraint I. Considering the allowed transitions, we have that
(28)$$P\left(s,s^{\prime }|I\right)=\left\{\begin{array}{c} \frac{1}{4}\;\mathrm{if}\;s\in \left\{s_{1},s_{4}\right\}\;\mathrm{and}\;s^{\prime }\in \left\{s_{1},s_{4}\right\},\\ 0\;\mathrm{otherwise}. \end{array}\right.$$

The matrix elements $p_{ij}=P\left(s=s_{i},s^{\prime }=s_{j}|I\right)$ are given in Table 1, together with the density matrix elements $\rho_{ij}$. The latter are simply
(29)$$\rho_{ij}=\frac{1}{2\eta }\left(p_{ij}+p_{ji}+i\;\left[p_{ij}-p_{ji}\right]\right)=\frac{p_{ij}}{\eta }=2p_{ij},$$
because $\eta =\sum_{i=1}^{4}p_{ii}=\frac{1}{2}$. The states given by Equation (27) are equivalent to the ket notation (|00〉,|11〉,|01〉,|10〉). The density matrix elements $\rho_{ij}$ correspond to the density operator of a pure, but entangled state,
(30)$${\hat{\rho }}_{AB}=\frac{1}{2}\left(|00\rangle \langle 00|+|00\rangle \langle 11|+|11\rangle \langle 00|+|11\rangle \langle 11|\right)=|\Phi^{+}\rangle \langle \Phi^{+}|$$
with $|\Phi^{+}\rangle$ as one of the Bell states,
(31)$$|\Phi^{+}\rangle :=\frac{1}{\sqrt{2}}\left(|00\rangle +\langle 11|\right).$$

The reduced density matrices are
$$\begin{array}{cc} \rho_{A} & =\frac{1}{2}\left(\begin{array}{cc} 1 & 0\\ 0 & 1 \end{array}\right),\\ \rho_{B} & =\frac{1}{2}\left(\begin{array}{cc} 1 & 0\\ 0 & 1 \end{array}\right). \end{array}$$

Hence, ${\hat{\rho }}_{AB}\neq {\hat{\rho }}_{A}⊗{\hat{\rho }}_{B}$, and the operator ${\hat{\rho }}_{AB}$ is not separable. The values of the von Neumann entropy are $S_{A}=S_{B}=\ln2$ and $S_{AB}$ = 0, the latter being a pure state; therefore, the system is subadditive. With this example, we have found a density operator that is not only correlated but could describe an emergent system, since the entropy of the whole is less than that of its parts.

## 7. Concluding Remarks

In this work, we reviewed the idea of emergence in the context of quantum mechanics and fragile systems in order to detect emergent behavior in systems under the framework of the density matrix theory. We proposed the mutual information as a descriptor of emergence in fragile systems [21]. When the parameter I>0, there is more information in the whole than its parts; therefore, new information emerges as a result of the interaction of the subsystems as correlations. We have presented a concrete example of an operator that could describe a subadditive system.

We believe that this formulation may give some insights about adaptative, self-sustainable and interconnected models. In particular, we aim to describe how, in an entangled system, the observer is part of the observation by being an inseparable element of it; for instance, by considering the side effect that produces the act of measurement (i.e., the *observer effect* [33,34,35]). 

## Figures and Tables

**Table 1 entropy-24-01676-t001:** The set of joint probabilities $p_{ij}=P\left(s=s_{i},s^{\prime }=s_{j}|I\right)$ (third column) for the fragile system arising from the two-integer example described in Section 6. The fourth column shows the values of the elements of the corresponding density matrix.

*i*	*j*	$p_{\mathrm{ij}}$	$\rho_{\mathrm{ij}}$
1	1	¼	½
1	2	0	0
1	3	0	0
1	4	¼	½
2	1	0	0
2	2	0	0
2	3	0	0
2	4	0	0
3	1	0	0
3	2	0	0
3	3	0	0
3	4	0	0
4	1	¼	½
4	2	0	0
4	3	0	0
4	4	¼	½

## Data Availability

Not applicable.

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
