# Peer review of "Quantum Mutual Information, Fragile Systems and Emergence"

_entropy, 2022, doi:10.3390/e24111676_

Round 1
Reviewer 1 Report
Navarrete & Davis provide a descriptor for (weak) emergence in terms of the additivity of the entropy of subsystems’ density matrices, which is captured by Eq. (19). This equation is limited, however, and merely tells whether there is an information deficit in two subsystems with respect to their combined system. Apart from this, the work contains little novelty, and the overall presentation is quite poor. Major revisions are deemed required before reconsidering this work for publication.
Specific comments:
The authors are overselling their work in the abstract. They do not “propose a quantitative definition of emergence” – although the second sentence is more realistic: “We propose the use of a descriptor based on the difference of von Neumann entropy…”
The ‘subadditivity property’ should be explained upon first use in line 28.
“Emergence is, as well, a remarkable representation of life phenomenon and the key aspect of how patterns can emerge from randomness.” This sentence (lines 48-49) is too vague and not further related to the text.
In the paragraph on weak emergence (lines 59-65), it is stated that “after a while the behavior can be unpredictable” without reference to chaos theory, although the latter imposes itself in this context.
Lines 97-98: “The main aspect of biological emergent systems is that they must be observed from inside them by the system.” This statement is either false or too vague to be true. What about observing bird flocking or the emergent behavior of ant colonies?
Where does the observation come from that “when we have a mixed density matrix the trace of the square density matrix is given by the inequality…” (lines 118-119)? Please motivate with references, and discuss the analogy between Eq. (10) and your result in Eq. (19), at least in the conclusions.
The exact meaning of rho_12 = rho_1 x rho_2 is not explained in line 132.
Lines 137-139: Why would Eq. (19) not hold for strong emergence as well?
Section 5: The meaning of E(s) is not explained in the text.
The concluding statement in lines 155-158 is very ambitious, and does not match what has been achieved in this work.
Technical:
Line 19: “as it was mentioned before” What does this refer to?
Lines 112-113: Rewrite “If all the vector are in the same state (meaning that the same probability distribution it is assigned)…”
Author Response
We have the response to the letter in the attached file.
Best regards.

Reviewer 2 Report
See File

Author Response

(The authors gave the same response as above.)

Round 2
Reviewer 1 Report
The new section on "Fragile systems" is helpful. Yet R_A lacks a definition; its meaning is not clear at this moment. Additionally, please provide a reference for the last sentence "As the modification of the state l depends on the details of the environment doing the measurement (which we do not know or control with accuracy), the outcome of a measurement is unavoidably stochastic, and a mathematical formulation requires probability theory."
Please at least rephrase the abstract and lines 184-187 to be more clear (use several sentences instead of one or two).
Author Response
Dear all,
Attached to you will find the letter with the responses.
Best regards.

Reviewer 2 Report
Well, I believe the manuscript has been sufficiently improved to warrant publication in Entropy.
Author Response
We thanks the reviewer for the helpful comments.
Best regards